# Bridging Modalities for Forgery Detection via Learnable Representations with Query-Guided Contrastive Learning

## Abstract

Image manipulation localization (IML) aims to identify tampered regions in edited images, which may range from object-level composites to subtle traces. Recent studies have began to explore the integration of multi-source cues, such as RGB, high frequency and noises, in pursuit of more precise localization. Despite this progress, the potential of cross-modal interactions and hierarchical perceptions deserves deeper investigation and exploitation. Inspired by how humans detect forgeries through dynamic zooming to capture holistic-local and semantic-detail cues, we propose BriQ (Bridge-Modality Query), a query-based framework that learns forged-aware representations to perceive multi-scale information. Meanwhile, we incorporate a structured attention to effectively model cross-modal interactions. To further enhance discriminative capability, we introduce query-to-regions contrastive learning (Q2R), which encourages representations to capture the essential contrast between tampered and authentic regions and aggregate forgery-related features, thereby significantly improving IML task performance. Extensive experiments conducted on multiple benchmark datasets validate BriQ's state-of-the-art effectiveness and robustness, while comprehensive ablation studies confirm the contributions of each component.

## 1 Introduction

Images have become essential evidence in modern life, shaping decision-making in domains ranging from journalism to justice. However, the ease of digital editing has led to a proliferation of manipulated images, crafted with increasingly sophisticated techniques. The computer vision community has responded to this challenge by exploring the task of Image Manipulation Localization (IML). Early methods adapt semantic segmentation methodologies, formulating forgery localization as a binary classification task based solely on RGB input to predict a pixel-level mask. These methods are effective in addressing manipulations with semantic inconsistencies. However, as tampering technologies evolve, forensic traces become increasingly faint, making pure RGB input inadequate for the accurate location of sophisticated manipulations.

To tackle this, recent studies have extended beyond RGB inputs, incorporating a broader spectrum of signals such as edge information, noise distributions, high-frequency cues, etc. These signals are often grouped under the "micro" perspective, complementing the "macro" view provided by semantic content. Research show that, compared to earlier RGB-only methods, combining both perspectives during training significantly enhances the detection of tampering and improves localization accuracy [Kwon et al. (2021); Guillaro et al. (2023); Zeng et al. (2024); Zhu et al. (2025b)]. A typical multi-modal IML pipeline comprises three stages: multi-source feature extraction, feature aggregation, and mask prediction. While notable progress has been made, current methods exhibit limitations both in establishing effective modality-interaction mechanisms and in precisely modeling intrinsic feature discrepancies between tampered and non-tampered regions.

First, during feature extraction, most approaches rely solely on single-level features to predict tampering and supervise the entire network. This oversight in architecture fails to leverage hierarchical features with varying receptive fields, neglecting the discriminative power inherent in multi-scale features for characterizing tamper artifacts. Additionally, in feature aggregation paradigms, macro-

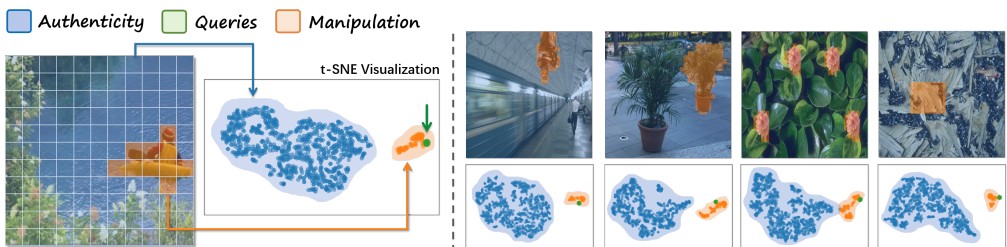

Figure 1: Manipulated examples and corresponding t-SNE projections of patch features and queries embeddings. Blue, orange and green denote authentic patches, forged patches, and forged-aware queries, respectively. The distinct clustering demonstrates BriQ's strong discriminative capability in identifying authentic and manipulated regions.

and micro-modality features are typically fused via naive operations such as concatenation, addition, or simple weighting. This simplistic approach overlooks the necessity of modality-aware interactions to capture cross-modal dependencies, which are essential for precise tamper localization. To overcome these limitations, we propose a novel query-based architecture that introduces a set of learnable tampering-aware representations. These queries serve as anchors to integrate hierarchical features and guide cross-modal interaction. Notably, such a design also mimics the human perceptual reasoning process, where observers alternate between global understanding and local inspection to detect inconsistencies.

Second, contrastive learning has emerged as a promising strategy in IML to enhance discrimination between manipulated and authentic content. Existing methods typically employ straightforward Region-to-Region(R2R) designs that aim to differentiate features between tampered and non-tampered regions. However, this approach faces limitations in homogeneous splicing tampering scenarios, where both regions may exhibit virtually indistinguishable appearances and statistical properties; thereby violating the core assumptions of R2R contrastive learning that requires separable feature distributions. To address this issue, we introduce a novel Query-to-Regions (Q2R) contrastive learning mechanism. Instead of contrasting regions directly, our approach formulates the objective as learning attractions and repulsions between different regions and the tampering representations, enabling to perceive tampering-specific features even when regional similarity is high. Ultimately, we enable the representations to become tampering-aware anchors, achieving differentiation and localization of manipulated informations, as shown in Fig. 1.

In summary, our contributions are three-fold:

- *Bidirectional Cross-modality Attention:* We present BriQ, a noval dual-modality, multi-level aggregation framework. BriQ leverages learnable tampering-aware representations to hierarchically propagate tampering signals, and introduces bidirectional attention mechanism to enable effective cross-modality interaction at each level.
- *Query-to-Regions Contrastive:* We introduce a Q2R contrastive learning that supervises queries to aggregate tampering-related cues while remaining distinct from authentic content. By aligning the internal reasoning agents directly with manipulated regions, this objective strengthens the model's sensitivity to subtle or ambiguous forgeries.
- *Superior Performance in Accuracy and Robustness:* We conduct extensive experiments to show that BriQ achieves state-of-the-art results on multiple datasets.

## 2 RELATED WORK

### 2.1 IMAGE MANIPULATION LOCALIZATION (IML)

Early approaches in IML mainly rely on RGB inputs and utilize single-branch CNN or Transformer architectures to achieve tampering localization, such as PSCC-Net and IML-ViT [Liu et al. (2022a); Ma et al. (2023)]. To handle increasingly sophisticated manipulation and perceive subtle tampering traces, recent studies have incorporated multiple signals (noise/frequency) and shifted toward dual-branch architectures for enhanced feature representation:

**CNN-based**: CAT-Net and ObjectFormer adopt two separate CNN encoders to extract RGB and frequency features, which are then aggregated through a decoder to identify tampering artifacts [Kwon et al. (2021); Wang et al. (2022)]. Similarly, MVSS-Net and MUN fuse features from RGB and noise branches via convolutional operations [Dong et al. (2022); Liu et al. (2025)].

**Transformer-based**: Compared to CNNs, ViT-based architectures exhibit stronger capabilities in modeling relationships between different regions [Ma et al. (2023)]. Trufor, MGQFormer and MMRL-Net employ dual Transformer encoders for RGB and noise maps [Guillaro et al. (2023); Zeng et al. (2024); Li et al. (2025)]. Trufor and MGQFormer merge features early via attention mechanisms, while MMRL-Net enforces consistency constraints at the output stage. Besides, FMAE [Zhu et al. (2025a)] combines three signals SRM, Bayar, Noiseprint++ and progressively injects micro-branch features into the RGB Transformer encoder to enrich its representations [Fridrich & Kodovsky (2012); Bayar & Stamm (2018); Guillaro et al. (2023)].

**Hybrid Architectures**: To strengthen model's ability of detecting both fine-grained traces and object-level manipulations, Mesorch adopts a hybrid CNN-Transformer architecture, where CNN processes fused RGB and high-frequency features, and Transformer handles the combination of RGB and low-frequency signals [Zhu et al. (2025b)] . The outputs across both branches (totaling 8 features) are aggregated through weighted combination for final localization.

## 2.2 LEARNABLE REPRESENTATIONS

Query-based Vision Transformer architectures employ learnable query embeddings as task-specific representations. By performing global attention over the entire image, these queries effectively capture holistic image information and have been widely adopted for specialized representations. DETR introduces object queries to detect the existence and localize targets, with each embedding encoding positional information to predict bounding boxes [Carion et al. (2020)]. MaskFormer, Mask2Former and AlignSeg utilize segment embeddings to represent latent objects for classification and localization, generating masks via cross-attention or dot-product operations [Cheng et al. (2021; 2022); Huang et al. (2021)]. BLIP2 bridges vision-language modalities, aligning text and image features by learnable queries in its Q-Former [Li et al. (2023)]. Adaptation to IML, MGQFormer pioneers the use of mask-guided forge and authentic class tokens [Zeng et al. (2024)]. These tokens interact with fused RGB-noise patch tokens through a self-attention decoder. However, these tampering tokens merely combine dual modalities, failing to explicitly model their inter-dependencies.

## 2.3 CONTRASTIVE LEARNING IN IML

To improve the discriminative power, contrastive learning has gained growing attention, owing to its inherent ability to model dichotomy problems without labeled data [van den Oord et al. (2019); Le-Khac et al. (2020)]. SAFIRE and MMRL-Net proposes region-to-region contrastive learning scheme that encourages consistency within the same source region while distinguishing between different sources [Kwon et al. (2024); Li et al. (2025)]. However, patch-based processing inherently leads to information blending in boundary patches containing mixed sources. This poses significant challenges for copy-move forgeries, where semantically identical (but provenance-distinct) regions resist effective separation through standard contrastive objectives. To mitigate this, NCL-IML employs dual projectors to map ambiguous patches to both authentic and forged feature spaces, then incorporates these projections as soft samples in the contrastive learning objective [Zhou et al. (2023)]. FOCAL clusters patch features from different sources into forged and authentic cluster centers, enforcing contrastive repulsion only between these cluster centroids [Wu et al. (2025)].

## 3 METHODOLOGY

We present BriQ, a query-based framework tailored for IML task. It is designed to explicitly model hierarchical interactions between global and high-frequency features through structured attention and contrastive learning. As shown in Fig. 2 and detailed in Algorithm 1, the framework consists of four main components: (1) a dual-stream feature extraction module for global and local clues; (2) a hierarchical bidirectional attention block enabling cross-modal interaction; (3) a novel query-to-different-regions contrastive alignment module for fine-grained supervision; and (4) a query-feature similarity voting mechanism that generates the final mask without decoder.

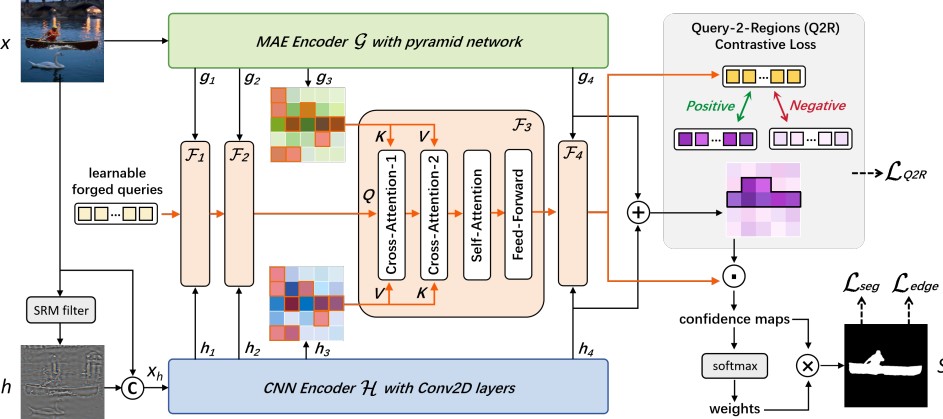

Figure 2: BriQ utilizes the original image and high-frequency information as two-modal inputs. During the modality aggregation stage $\mathcal{F}$, learnable queries interact alternately with hierarchical features from both modalities. Subsequently, Q2R contrastive learning is applied to pull the queries closer to forged patches while pushing them away from authentic ones. The final forgery map $S$ is obtained by adopting a lightweight voting mechanism.

## 3.1 DUAL-STREAM FEATURE EXTRACTION

Image forgeries often manifest as either global inconsistencies, such as contextually implausible objects, or local artifacts, including abruptive edge transitions or texture anomalies. To comprehensively capture these cues, we design a dual-stream encoder consisting of two specialized branches: a global branch for object-level features and a local branch for fine-grained patterns.

We begin with the extraction of local manipulation-sensitive features like Mesorch. Following prior work on forensic signals, we apply the SRM filter to the original image $x \in \mathbb{R}^{3 \times h \times w}$ to obtain high-frequency features $h \in \mathbb{R}^{3 \times h \times w}$, which highlight traces such as texture abnormalities. These residuals are concatenated with the original image to form a six-channel input $x_h \in \mathbb{R}^{6 \times h \times w}$, which is then fed into a ConvNeXt-Tiny encoder [Liu et al. (2022b)]. The encoder $\mathcal{H}$ extracts hierarchical feature representations at four resolution stages, denoted as $\{\hat{h}_1, \hat{h}_2, \hat{h}_3, \hat{h}_4\}$, where $\hat{h}_i \in \mathbb{R}^{(48 \cdot 2^{5-i}) \times \frac{h}{2^{6-i}} \times \frac{w}{2^{6-i}}}$, and each stage comprises multiple residual blocks. We then unify the channel dimensions of all feature maps via four convolutions (Conv2D) with fixed output channels, ensuring compatibility between different modalities. The encoder outputs hierarchical feature maps: $Proj(\mathcal{H}(x_h)) = \{h_1, h_2, h_3, h_4\}$, where $h_i \in \mathbb{R}^{256 \times \frac{h}{2^{6-i}} \times \frac{w}{2^{6-i}}}$, spanning from semantically rich deep features to spatially precise shallow features.

In parallel, we construct a global branch to encode object-level semantics and contextual relations. Specifically, we adopt a masked autoencoder(MAE-Base) as the backbone, augmented with a Feature Pyramid Network(FPN) to extract sufficient multi-scale features [He et al. (2021); Ma et al. (2023); Li et al. (2022)]. The encoder $\mathcal{G}$ extracts the patch embeddings of the last block as coarse global information $\hat{g} \in \mathbb{R}^{768 \times \frac{h}{16} \times \frac{w}{16}}$. Then, using a 4-layer FPN, we obtain multi-scale feature maps with the same channel dimensions, structurally aligned with the local stream. The global stream outputs four global representation: $FPN(\mathcal{G}(x)) = \{g_1, g_2, g_3, g_4\}$, where $g_i \in \mathbb{R}^{256 \times \frac{h}{2^{6-i}} \times \frac{w}{2^{6-i}}}$, with $g_1$ corresponding to the coarsest semantic feature with the largest receptive field, and $g_4$ preserving the finest spatial resolution. This parallel design ensures that both abstract and forensic cues can be accessed and reasoned over in a layer-aligned manner in subsequent modules.

## 3.2 CROSS-MODALITY WITH PATCH-ALIGNED ATTENTION

We introduce a set of learnable representations $q \in \mathbb{R}^{N_q \times d}$, where $N_q$ denotes the number of embeddings and $d$ the dimension. The proposed embeddings hierarchically refine representations of potential forged regions through bidirectional cross-modal module $\mathcal{F}$. Specifically, the mechanism first inspects global context to identify objective anomalies, then examines fine details to detect localized artifacts, or vice versa.

Early methods aggregate multi-modal features via concatenation ($w_1 x + w_2 x_h$) or feature-level addition ($\sum_i w_i g_i + \sum_j w_j h_j$), where gradients are propagated only through the fusion weights $w_{1,2}(x, x_h)$ or $w_{ij}(g_i, h_j)$. This limits the optimization of direct cross-modal correlation. To address this, our method introduces explicit interaction at the attention level. In our method, for each hierarchy level $i \in \{1, 2, 3, 4\}$, module $\mathcal{F}_i$ contains a two-step cross-attention process $\mathcal{CA}$, followed by self-attention $\mathcal{SA}$ and feed-forward layers $\mathcal{FFN}$. The placement of $\mathcal{SA}$ after $\mathcal{CA}$ is inspired by Mask2Former, where late-stage $\mathcal{SA}$ enhances cross-modal knowledge acquired during $\mathcal{CA}$ and intra-query consistency [Cheng et al. (2022)]. The update rule is defined by lines 4-8 of Algorithm 1.

---

**Algorithm 1** Algorithm Flow for BriQ

---

**Require:** Input data $x$, ground truth mask $M$
**Ensure:** Final loss $\mathcal{L}$
1: Initialize $q_0$
2: Extract features $g_i$, $h_i$ for $i = 1$ to 4:

$$\{g_1, g_2, g_3, g_4\} = FPN(\mathcal{G}(x)) \quad (1)$$
$$\{h_1, h_2, h_3, h_4\} = Proj(\mathcal{H}([x, h])) \quad (2)$$

3: Hierarchical feature aggregation:
4: **for** $i = 1$ to 4 **do**
5: $\quad q_i \leftarrow \mathcal{CA}_i^1(q_{i-1}, g_i, h_i)$
6: $\quad q_i \leftarrow \mathcal{CA}_i^2(q_i, h_i, g_i)$
7: $\quad q_i \leftarrow \mathcal{FFN}_i(\mathcal{SA}_i(q_i))$
8: **end for**
9: Compute Score map $S$
10: Compute $\mathcal{L}_{Q2R}$, $\mathcal{L}_{seg}$ and $\mathcal{L}_{edge}$
11: Return the final loss $\mathcal{L}$

---

Here, $\mathcal{CA}(q, k, v)$ denotes our cross-modal attention module, where queries $q$ attend to modality $k$ by computing attention weights and aggregate information from modality $v$ to achieve the modality-aware feature fusion. Formally:

$$\mathcal{CA}(q, k, v) = q + A(q, k, v) = q + W_v v \, \text{Softmax}\left(\frac{q^T W_q^T W_k k}{\sqrt{d_k}}\right). \quad (3)$$

This bidirectional structure, where $g$ and $h$ alternatively act as $k$ and $v$ across two successive attention stages, explicitly embeds interactions between global and local modalities, which is absent in uni-modal attention, such as $\mathcal{CA}(q, g, g)$ or $\mathcal{CA}(q, h, h)$.

**Theoretical Analysis**. We perform gradient flow analysis on attention output $A(q, g, h)$, examining gradients w.r.t. queries $q$, global info $g$, and high-frequent $h$ to validate our design.

$$\text{(a)} \quad \frac{\partial vec(A)}{\partial vec(q)} = (I \otimes W_v \boxed{h}) \frac{J}{\sqrt{d_k}}(\boxed{g}^T W_k^T W_q \otimes I)$$

$$\text{(b)} \quad \frac{\partial vec(A)}{\partial vec(g)} = (I \otimes W_v \boxed{h}) \frac{J}{\sqrt{d_k}}(I \otimes q^T W_q^T W_k) \quad (4)$$

$$\text{(c)} \quad \frac{\partial vec(A)}{\partial vec(h)} = Softmax(...)^T \otimes W_v,$$

where $J$ is a block diagonal matrix related to $A$ and $W_q, W_k, W_v$ are linear weight matrices.

In equation (4a), the gradient of $\frac{\partial vec(A)}{\partial vec(q)}$ depends jointly on both $g$ and $h$, indicating that the query token receives feedback influenced by both modalities. Similarly, the gradient with respect to $g$ involves $h$, indicating that modality $h$ directly influences $g$. This highlights the cross-modal interaction, suggesting that $g$ also benefits from direct dual modal supervision, as demonstrated in equation (4b). However, equation (4c) reveals that $h$'s gradient depends solely on the Softmax of relation between $q$ and $g$ without directly incorporating signals from $g$ or $q$. To address this, our bidirectional structure $(q, g, h)(q, h, g)$ in Eq.(3) enables mutual participation of $g$ and $h$ in query refinement. This contrasts with conventional fusion strategies or uni-modal attention, propagating solely through weights or queries, lacking such direct interaction mechanisms. It can be further verified by analogy to Equation (4).

Beyond structural interactions, our framework implements a hierarchical coarse-to-fine pipeline. At lower levels (e.g., $i = 4$), where feature maps preserve finer spatial details, the queries attend to dense patch tokens for precise localization of subtle tampering traces. This framework overcomes the quadratic complexity limitation $O(N^2)$ (where $N = H/P \times W/P$ is the number of patch tokens), inherent in standard transformer architectures, as our query-based cross-attention maintains efficient $O(N_q \cdot N)$ scaling, where $N_q < N$, supporting high-resolution reasoning. To further

enhance efficiency and robustness, we randomly partition $g$ and $h$ into two spatial halves while preserving patch alignments: $\mathcal{CA}(q, g, h) = \mathcal{CA}(\mathcal{CA}(q, g^+, h^+), g^-, h^-)$, where $g^+$ and $h^+$ represent one half of the partitions, while $g^-$ and $h^-$ represent the complementary halves.

In summary, this module introduces a novel hierarchical and bidirectional cross-modal attention mechanism. It aligns semantic and forensic clues across scales, structurally embeds their interaction into query refinement, and supports efficient, interpretable, and fine-grained forgery localization.

### 3.3 QUERY-TO-REGIONS CONTRASTIVE LEARNING

To enhance the discriminative power of our learnable queries and ensure they accurately represent forged region patterns, we introduce the first Query-to-Region (Q2R) contrastive learning strategy in IML. This approach fundamentally differs from conventional Region-to-Region (R2R) paradigms.

A set-metric inequality inspired Q2R design. The objective of R2R is to maximize the distance $d(\mathcal{A}, \mathcal{F})$ between the two patch sets (authentic $\mathcal{A}$ and forged $\mathcal{F}$). Q2R introduces a third query set $\mathcal{Q}$, treats set $\mathcal{F}$ (tampered) as positive samples, and set $\mathcal{A}$ (authentic) as negative samples. Its objective is to minimize the distance $d(\mathcal{Q}, \mathcal{F})$ (query to tampered) while maximizing $d(\mathcal{Q}, \mathcal{A})$ (query to authentic). According to the inequality $d(\mathcal{Q}, \mathcal{F}) > d(\mathcal{A}, \mathcal{F}) - d(\mathcal{Q}, \mathcal{A}) > 0$, as training progresses,$d(\mathcal{A}, \mathcal{F})$ increases and $d(\mathcal{A}, \mathcal{F})$ approaches $d(\mathcal{Q}, \mathcal{A})$. This design avoids the need for a direct partition between $\mathcal{A}$ and $\mathcal{F}$ inherent in R2R. This approach maintains effective region discrimination while simplifying the learning process. By leveraging the intermediate queries, Q2R ensures that the queries can still capture and differentiate the subtle features of tampered regions, thereby enhancing the overall robustness and efficiency of the model.

To construct the Q2R objective, we first divide the lowest-level features (i.e., $g_4$ or $h_4$) into patches aligned with the ground truth mask. A patch is labeled as positive if more than 25% are tampered while patches with no tampered pixels are treated as negative. We then define the contrastive loss via the InfoNCE objective:

$$\mathcal{L}_{Q2R} = \sum_{n \in [1, N_q]} InfoNCE(q_n, f_e, a_e) = \sum_{n \in [1, N_q]} -E[\log \frac{\sum_f \langle q_n, f_e \rangle}{\sum_f \langle q_n, f_e \rangle + \sum_a \langle q_n, a_e \rangle}], \quad (5)$$

where $f_e$ and $a_e$ are feature embeddings from forged and authentic patches, respectively.

### 3.4 FINAL MASK PREDICTION

To generate the final tampering mask, we adopt a lightweight prediction mechanism based on query-to-feature similarity voting. Specifically, we first fuse the lowest global and local features by simple averaging: $p_4 = \frac{1}{2}(g_4 + h_4)$. Each learnable query $q_n$ then performs a dot product with every patch embedding in $p_4$, producing a set of query-to-feature confidence maps. These maps are multiplied by the weights obtained through softmax, and then summed to get the final prediction result $S$, specifically $S = \sum_{n=1}^{N_q} (\text{Softmax}(\langle q_n, p_4 \rangle) \times (\langle q_n p_4 \rangle))$. The prediction map $S$ is supervised by both segmentation and edge-aware objectives: (1) $\mathcal{L}_{seg}$ compares the predicted tampering map $S$ against the ground truth binary mask $M$ using standard Binary Cross-Entropy: $\mathcal{L}_{seg} = BCE(S, M)$. (2) $\mathcal{L}_{edge} = BCE(S, M, weight = M_{edge})$ enhances edge supervision, follow IML-ViT. The final combined loss is formulated as $\mathcal{L} = \mathcal{L}_{seg} + \lambda_1 \mathcal{L}_{edge} + \lambda_2 \mathcal{L}_{Q2R}$, where $\lambda_1 = 20$, $\lambda_2 = 0.1$ are served as normalized weights to account for the differing gradient scales.

## 4 EXPERIMENTS

We conduct comprehensive experiments to evaluate BriQ on a standard IMDLBenCo benchmark [Ma et al. (2024)]. This section is organized as follows: (1) first describe the experimental settings, including datasets, baselines, evaluation metrics, and implementation details; (2) then present quantitative and qualitative comparisons with state-of-the-art methods; (3) finally conduct ablation studies to analyze the effectness of designed components in our framework.

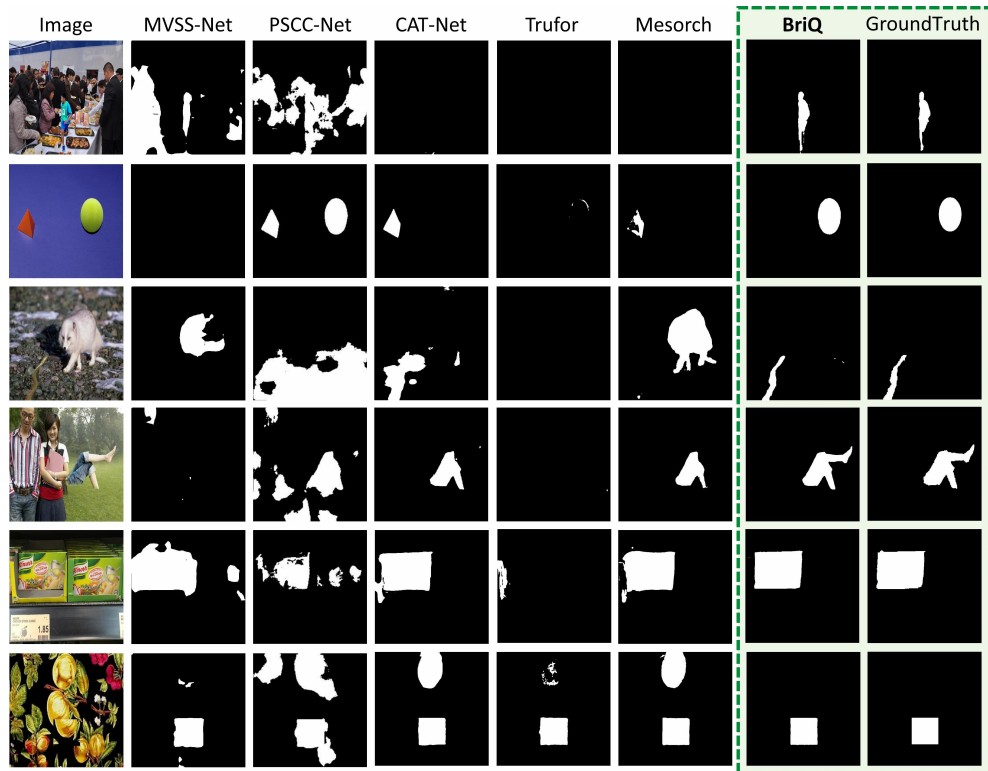

Figure 3: Qualitative comparison with state-of-the-art IML methods.

Table 1: Quantitative comparison with state-of-the-art IML methods. Avg. represents the average score on test sets. Best and second-best results are highlighted in bold and underlined, respectively.

| Method | F1↑ | | | | | Permute-F1↑ | | | | |
|---|---|---|---|---|---|---|---|---|---|---|
| | Coverage | Columbia | NIST16 | CASIAv1 | Avg. | Coverage | Columbia | NIST16 | CASIAv1 | Avg. |
| MVSS-Net | .4860 | .7399 | .3363 | .5832 | .5364 | .5172 | .7879 | .3775 | .6016 | .5711 |
| PSCC-Net | .4475 | .8841 | .3457 | .6304 | .5769 | .4930 | .8937 | .3944 | .6382 | .6048 |
| CAT-Net | .4273 | **.9150** | .2521 | .8081 | .6006 | .5165 | .9547 | .3316 | .8154 | .6546 |
| TruFor | .4573 | .8845 | .3480 | .8176 | .6269 | .5369 | .9547 | .4046 | .8340 | .6826 |
| Mesorch | .5862 | .8903 | .3921 | .8398 | .6771 | .6346 | **.9708** | .4514 | .8472 | .7259 |
| **BriQ** | **.6976** | .8972 | **.5199** | **.8549** | **.7424** | **.7189** | .9637 | **.5495** | **.8599** | **.7730** |

## 4.1 EXPERIMENTAL SETUP

**Datasets.** We follow the standard Protocol-CAT, a widely adopted training protocol in IML [Kwon et al. (2022)]. The training set contains five public datasets, including CASIAv2, Fantastic Reality, IMD2020, tampered COCO and tampered RAISE, with fixed-size sampling from each source in every epoch [Dong et al. (2013); Kniaz et al. (2019); Novozamsky et al. (2020); Kwon et al. (2022)]. These datasets incorporates both classical forgeries (e.g. splicing, blurring, compression) and advanced editing (e.g. copy-move and cross-image composition), covering a broad manipulation spectrum. Evaluation is conducted on four widely used test sets to assess generalization across manipulation types and domains: CASIAv1, Coverage, NIST16, and Columbia [Dong et al. (2013); Wen et al. (2016); Guan et al. (2019); Hsu & Chang (2006)].

**Baselines.** We compare BriQ against state-of-the-art IML methods: MVSS-Net, PSCC-Net, Cat-Net, TruFor and Mesorch. To ensure fair comparison, we adopt IML-Benco, a standardized benchmark offering unified data loaders, testing pipelines, and evaluation metrics.[Chen et al. (2021); Liu et al. (2022a); Kwon et al. (2022); Guillaro et al. (2023); Zhu et al. (2025b)]

| Method | Avg. F1↑ | | |
|---|---|---|---|
| | GN | GB | JC |
| MVSS-Net | .5744 | .2962 | .5061 |
| PSCC-Net | .5639 | .3282 | .4925 |
| CAT-Net | .7802 | .5312 | .7352 |
| TruFor | .7286 | .5320 | .7049 |
| Mesorch | .7998 | .6016 | **.7738** |
| **BriQ** | **.8301** | **.6719** | .7734 |

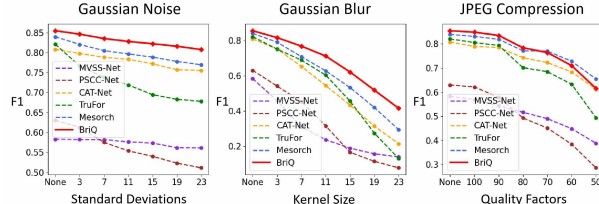

Table 2: Robustness test: Average F1 score on CASIAv1 under various perturbations. GN, GB, and JC represent Gaussian Noise, Gaussian Blur, and JPEG Compression, respectively.

Figure 4: Robustness test: F1 score on CASIAv1 under increasing level of various perturbations. The red solid line represents the performance of BriQ, demonstrating superior robustness compared to others.

**Metrics.** Following the public benchmark, we report two pixel-level evaluation metrics: F1-score, computed at a 0.5 threshold to evaluate localization performance; Permute-F1, defined as $\max(F1, Inverted\text{-}F1)$, serves to evaluate the model's predictive ability to distinguish between tampered and non-tampered regions.

**Implementation Details.** In training process, images are uniformly resized to 512×512, with both standard augmentations (flip, rotation, brightness) and IML-specific methods (random inpainting, copy-move). BriQ is trained for 150 epochs with batch size 4 on 8 NVIDIA V100 GPUs. We use AdamW optimizer with a learning rate of 1e-4, weight decay of 0.05, and cosine annealing. A 4-epoch warm-up precedes cosine decay to 5e-7.

## 4.2 PERFORMANCE EVALUATION

**Localization.** Tab. 1 reports the quantitative results on four test sets. BriQ consistently achieves the best results, outperforming almost all prior methods across F1 and Permute-F1 metrics. Notably, BriQ delivers an average improvement of +6.53% in F1 and +4.71% in Permute-F1 over the second-best method, demonstrating its strong performance. As illustrated in Fig. 3, BriQ produces sharper and more accurate tampering masks, especially at small-scale and imperceptible manipulations. These qualitative improvements align with the quantitative gains and highlight the benefit of our cross-modality reasoning and query-based contrastive learning mechanism.

**Robustness.** We evaluate the robustness of all methods under three most common image corruptions: Gaussian Noise, Gaussian Blur, and JPEG Compression, each applied at increasing intensity levels. The results are shown in Tab. 2 and Fig. 4. BriQ demonstrates remarkable resilience, outperforming others under noise and blur with margins of +3.03% and +7.03% in Avg. F1, respectively.

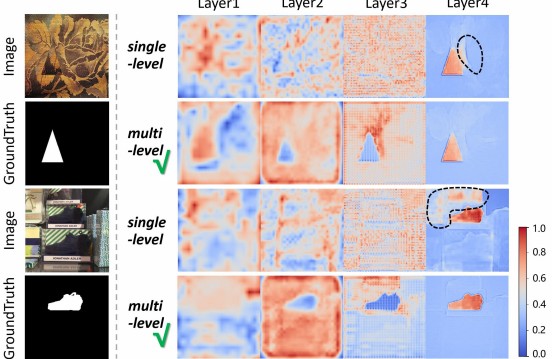

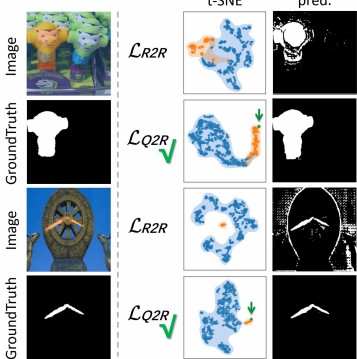

Figure 5: Similarity between the feature of each layer and queries under multi-level and single-level attention. Multi-scale information can help distinguish between manipulated and authentic features.

Figure 6: Feature distribution of queries and patch embeddings under Q2R paradigm and R2R paradigm. Forged patches are in orange, and authentic patches are in blue.

Table 3: Ablation studies on integration strategy, contrastive learning scheme, and query quantity.

| Setting | Details | Avg. F1↑ | Avg. P-F1↑ |
|---|---|---|---|
| **BriQ** | multi-level + $\mathcal{F}$ + $\mathcal{L}_{Q2R}$ + 16-query | **.7424** | **.7730** |
| w/o multi-level | $\mathcal{F}_4$ only | .7271 | .7575 |
| w/o $\mathcal{F}$ | $(q, h, g)(q, g, h)$ | .7249 | .7562 |
| | $(q, g, g)(q, h, h)$ | .7334 | .7620 |
| | $(q, h, h)(q, g, g)$ | .7334 | .7671 |
| | $(q, [g, h], [g, h])$ | .7258 | .7554 |
| w/o $\mathcal{L}_{Q2R}$ | $\mathcal{L}_{R2R}$ | .6979 | .7280 |
| #queries | 4-query | .7313 | .7620 |
| | 8-query | .7329 | .7635 |
| | 32-query | .7186 | .7469 |

Under JPEG compression, BriQ performs best at light compression and remains highly competitive at stronger levels, ranking second with a mere 0.04% gap in average performance.

## 4.3 ABLATION STUDY

We conduct detailed ablation studies to validate the contributions of BriQ's components, including integration strategy, contrastive learning design, and query quantity.

**Hierarchical Strategy.** To validate the effectiveness of hierarchical integration, we replace it with a single-level approach, which performs modality aggregation solely at the shallowest layer $\mathcal{F}_4$. As shown in Tab. 3 and Fig. 5, compared to single-level attention, employing multi-level feature aggregation allows the queries to effectively capture the distinctions between tampered and authentic features at each layer of the model, thereby enabling more precise localization of forgery regions.

**Attention across modality.** We conduct comprehensive ablation studies on cross-modal attention mechanisms. For conciseness, we represent the QKV attention mechanism as triplets. Our proposed bidirectional cross-attention first uses global features as Keys and high-frequency information as Values, then swaps their order, denoted as $(q, g, h)(q, h, g)$. We first compare it with per-modality attention strategies executed sequentially: $(q, g, g)(q, h, h)$ and $(q, h, h)(q, g, g)$. Tab. 3 shows performance degradation when cross-modal interaction is absent. Second, we modify the Key selection strategy, adopting $(q, h, g)(q, g, h)$, experimentally validating our design choice of using global features as primary Keys. Finally, we compare against the conventional concatenation fusion $(q, [g, h], [g, h])$. Through gradient analysis and experimental result, we demonstrate our strategy outperforms this weighted fusion approach in effectiveness.

**Contrastive Learning Design.** We replace contrastive supervision $\mathcal{L}_{Q2R}$ with $\mathcal{L}_{R2R}$, which directly drives a separation between the feature distributions of the two regions. As shown in Fig. 6, $\mathcal{L}_{Q2R}$ effectively discriminates the distributions of forged and authentic patches, while the queries exhibiting proximity to the forged ones. The corresponding result in Tab. 3 demonstrates that $\mathcal{L}_{Q2R}$ significantly outperforms $\mathcal{L}_{R2R}$ in forgery localization, highlighting its superior capability.

**Query Quantity.** As shown in Tab. 3, changing the query count to 4, 8, or 32 results in performance drops, and 16 queries are suitable for capturing diverse tampering patterns.

## 5 CONCLUSION AND FUTURE WORK

In this paper, we propose a novel structured framework for image manipulation localization, built upon learnable forged representations that evolves across feature maps from multi-scale modalities. Our method demonstrates strong performance on challenging benchmarks while providing new possibilities for structural understanding in IML. In future work, we plan to extend this framework to broader domains, including generative images produced by diffusion-based techniques. Besides, large language models can be integrated to enhance interpretability and deliver accessible natural language explanations.

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

## 6 APPENDIX

We present **BriQ**, a query-based framework tailored for Image Manipulation Location (IML) task. Our method unifies and extends prior efforts in query-based modeling, cross-modal fusion, and contrastive supervision under a structured and interpretable framework.

We build upon the idea of using learnable query tokens, but go beyond by explicitly modeling their evolution across multiple feature levels, enabling hierarchical interaction. Unlike previous approaches where queries passively attend to features, our query token actively performs alternating attention with global and local features at each level, forming a multi-stage reasoning path that mirrors human perception.

To further enhance discrimination, we introduce a query-to-regions (Q2R) contrastive loss that supervises the query token itself, guiding it to aggregate forged-region features while repelling pristine ones. Compared with prior work where contrastive losses are only applied globally, our design integrates contrastive supervision into the query refinement loop, aligning feature learning with the model's internal inference trajectory.

In this way, our approach not only inherits the strengths of prior models which contains multi-scale fusion, semantic-noise complementarity, and discriminative learning, but also introduces a layer-aware, interpretable reasoning structure tailored for fine-grained manipulation localization.

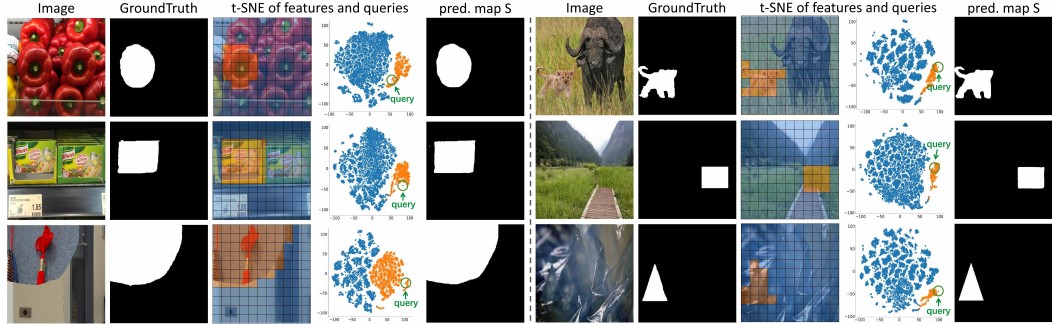

Figure 7: More manipulated examples and corresponding t-SNE projections of features and queries embeddings. Blue, orange and green denote authentic patches, forged patches, and forged-aware queries, respectively.

## 6.1 Feature distribution

After the training process is complete, the learnable representations evolve into tampering-aware anchors, achieving differentiation and localization of manipulated information. As illustrated in Fig.7, the BriQ model demonstrates a clear separation between the feature distributions of forged (orange) and authentic (blue) data. The query tokens (green), having been trained to focus on these distinctions, naturally gravitate towards the forged features in the feature space. This proximity indicates that the query tokens effectively capturing the essence of the manipulated information, enabling the model to perform detailed analysis and detection of tampering. This capability is essential for applications requiring high accuracy in identifying and localizing manipulated content.

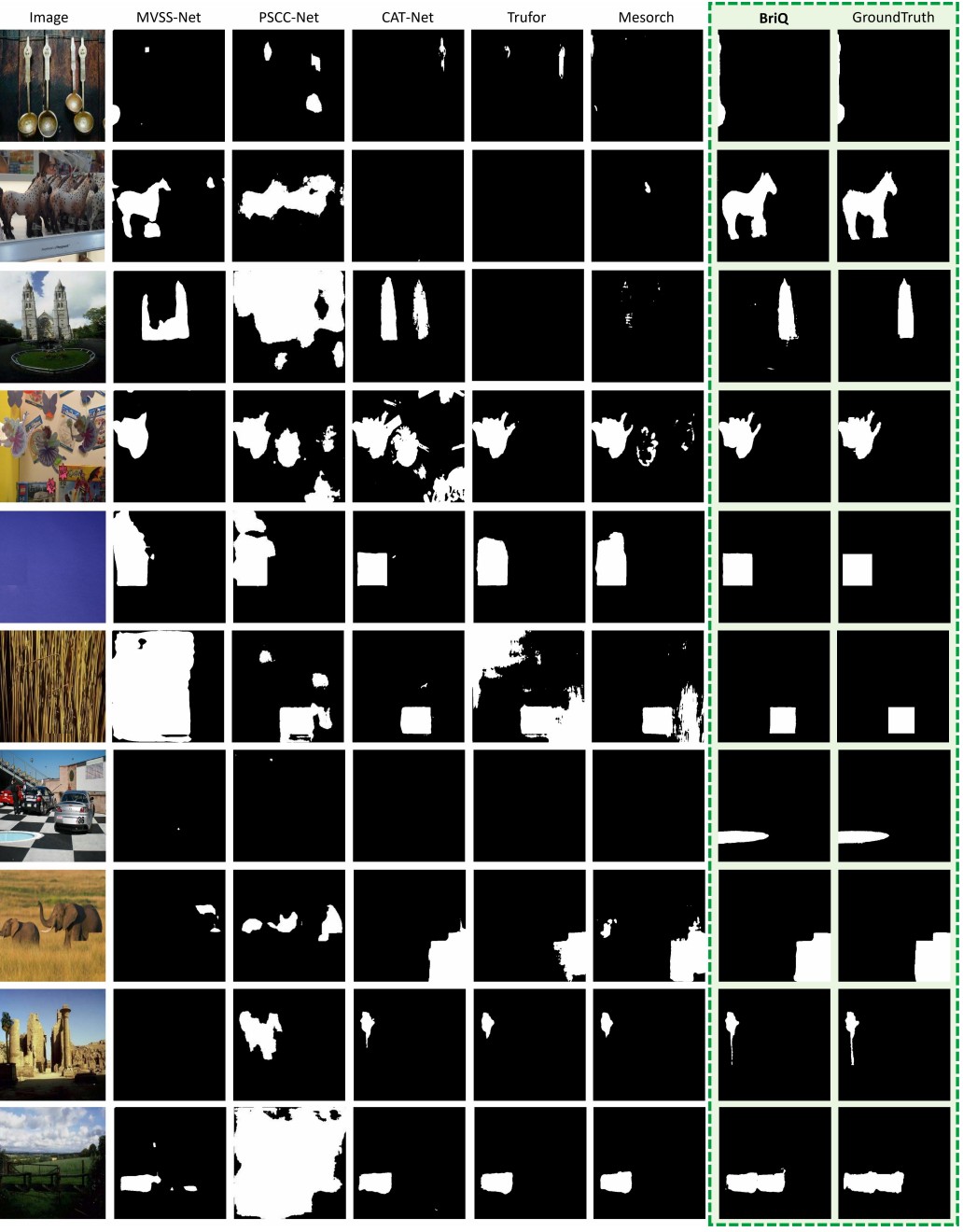

Figure 8: More qualitative comparisons-1.

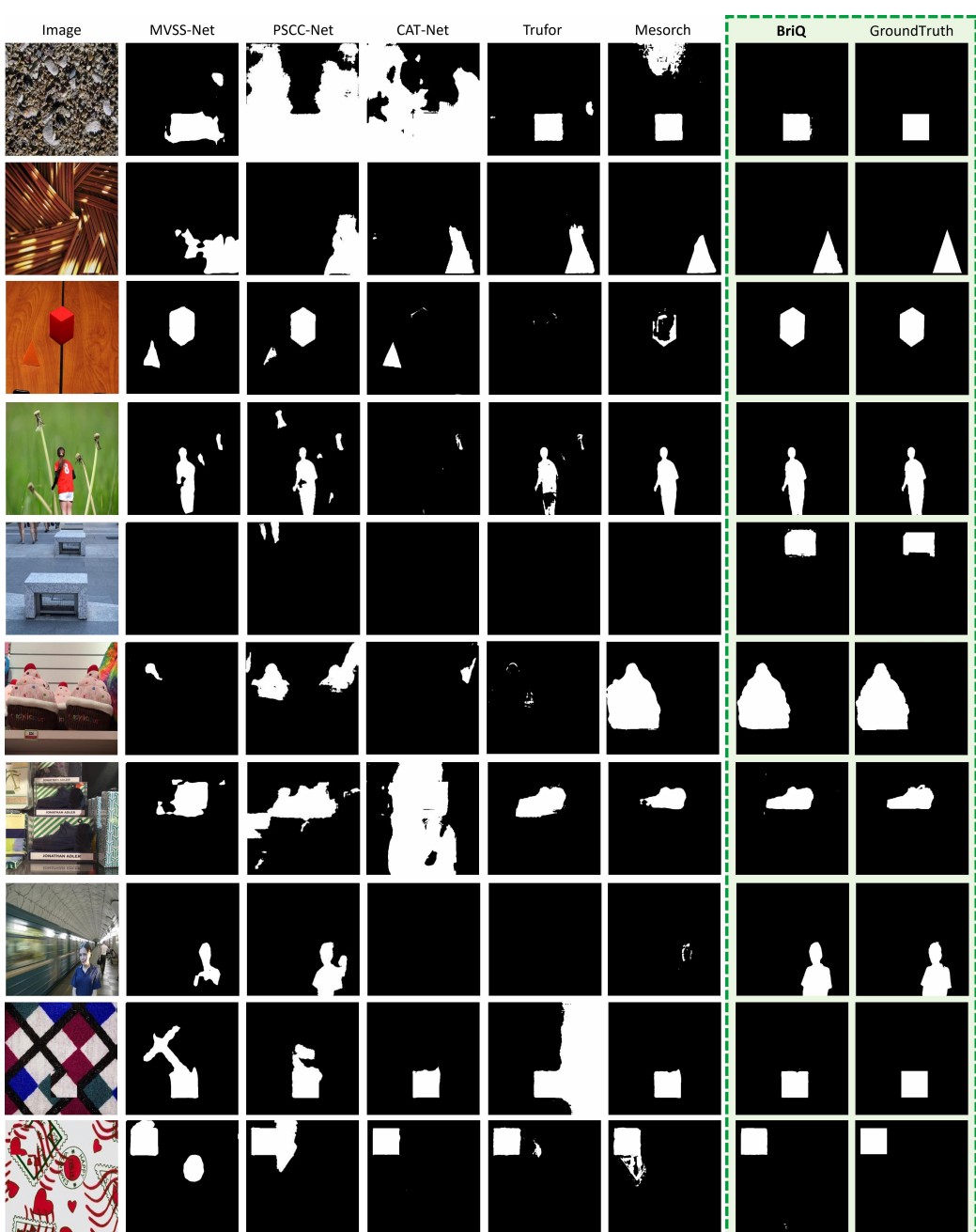

Figure 9: More qualitative comparisons-2.

## 6.2 LOCALIZATION PERFORMANCE

### 6.2.1 MORE QUALITATIVE COMPARISON

BriQ achieves high localization accuracy across a diverse range of manipulation techniques, such as duplication, cross-image collaging, and perceptually challenging subtle tampering operations. More comparisons in Fig.8 and Fig.9 showcase BriQ's ability to accurately pinpoint the locations of manipulations, regardless of their complexity or subtlety.

Table 4: Quantitative comparison with SparseViT.

| Method | F1↑ | | | | | Permute-F1↑ | | | | |
|---|---|---|---|---|---|---|---|---|---|---|
| | Coverage | Columbia | NIST16 | CASIAv1 | Avg. | Coverage | Columbia | NIST16 | CASIAv1 | Avg. |
| SparseViT | .5164 | **.9567** | .3726 | .8196 | .6663 | .5832 | **.9746** | .4292 | .8308 | .7045 |
| **BriQ** | **.6976** | .8972 | **.5199** | **.8549** | **.7424** | **.7189** | .9637 | **.5495** | **.8599** | **.7730** |

### 6.2.2 MORE QUANTITATIVE COMPARISON

We supplement the quantitative metrics of SparseViT [Su et al. (2025)] under the same training conditions. As shown in Tab. 4, BriQ achieves more favorable results.

## 6.3 AIGC BENCHMARK

To verify the effectiveness of BriQ on generative forgery, we adopt AIGC-Editing manipulation dataset [1] and supplement the training and test sets to the existing dataset while keeping other configurations unchanged. We select two recent state-of-the-art methods, Mesorch and SparseViT, and train them following their scripts. The quantitative results for each test set are shown in Tab.5, which illustrates the broad effectiveness of BriQ in addressing various tampering types—encompassing both traditional tampering and generative forgery. For generative forgery, the qualitative comparison is shown in Fig.10, while more of our visualization results are presented in Fig.11.

Table 5: Quantitative comparison on various test set, including AIGC test set.

| Method | F1↑ | | | | | | Permute-F1↑1 | | | | | |
|---|---|---|---|---|---|---|---|---|---|---|---|---|
| | Coverage | Columbia | NIST16 | CASIAv1 | AIGC | Avg. | Coverage | Columbia | NIST16 | CASIAv1 | AIGC | Avg. |
| Mesorch | .5726 | .9113 | .3786 | .8404 | **.8788** | .7163 | .6286 | **.9760** | .4386 | .8501 | .8833 | .7553 |
| SparseViT | .5554 | **.9137** | .2502 | .7734 | .7955 | .6576 | .6176 | .9587 | .3238 | .7894 | .8138 | .7007 |
| **BriQ** | **.6579** | .8938 | **.4864** | **.8541** | .8699 | **.7524** | **.6838** | .9572 | **.5238** | **.8592** | **.9731** | **.7994** |

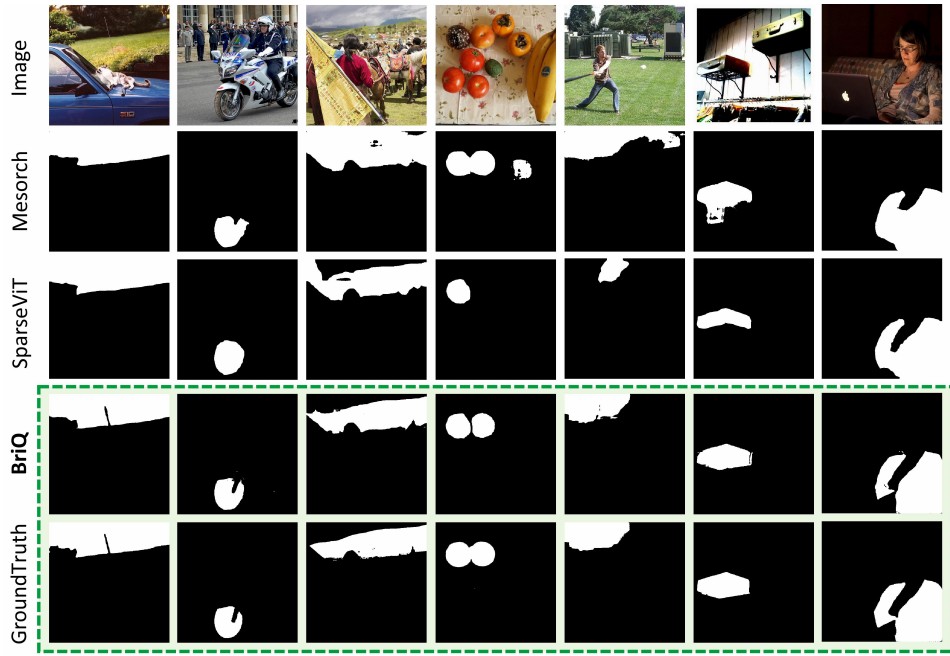

Figure 10: Qualitative comparison on AIGC test set.

---

[1] https://huggingface.co/datasets/zhipeixu/SD_inpaint_dataset

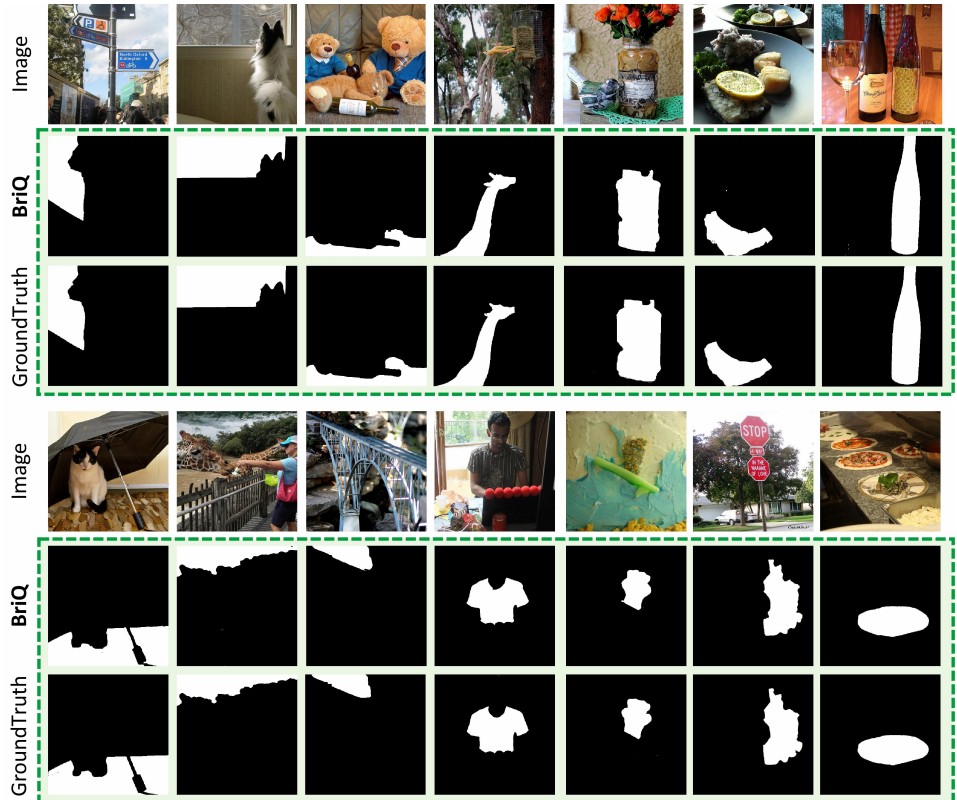

Figure 11: More visualization results of BriQ on AIGC test set.

## 6.4 ATTENTION MAP

We visualize the attention map between queries and macro features first, followed by micro features at each layer. Multi-head attention is aggregated via averaging, producing a $(bs, q, hw)$ attention map. After applying mean pooling along the query dimension, we obtain the attention between queries and a single modality, which is visualized as a heatmap.

As shown in Fig.12, for each image, our dual-branch architecture extracts its RGB features (top row) and high-frequency features (bottom row) across four layers (left to right columns). Following the arrow directions: first, the tampering query computes attention with deep RGB features, but this attention applies to deep high-frequency features; then reversely, the query computes attention with deep high-frequency features that affects RGB features. This completes one layer's computation before proceeding to the next layer.

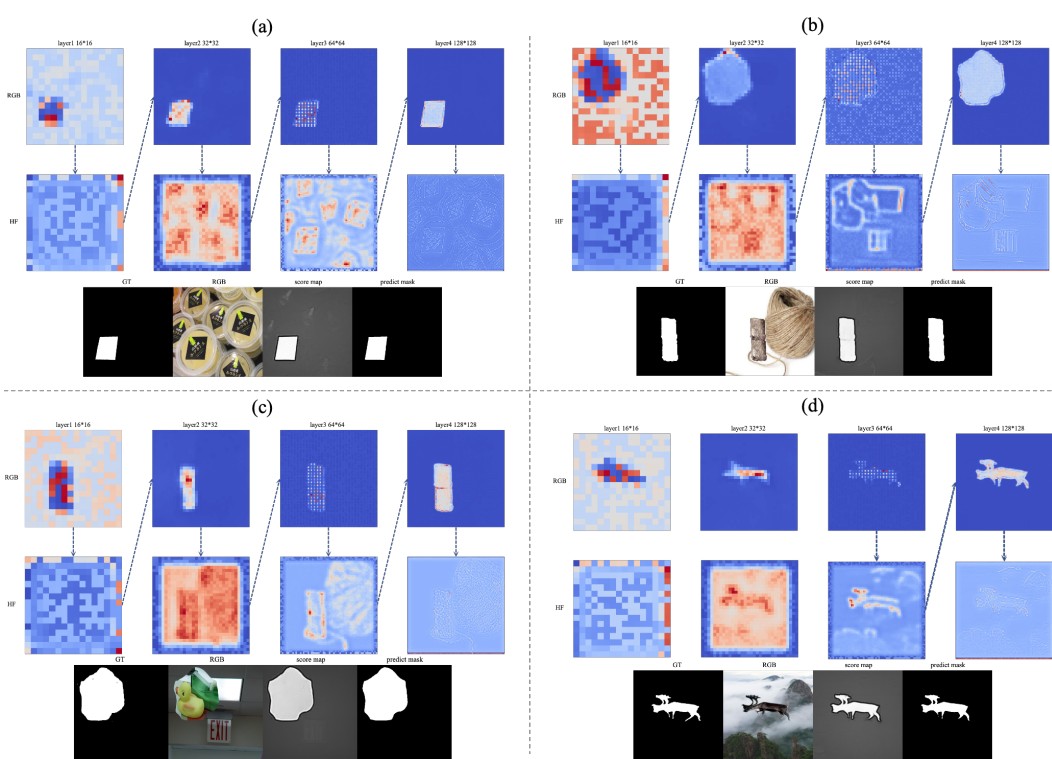

Figure 12: Visualization of the attention map between queries and features from dual modalities in BriQ.

