# OpenReview forum: "Bridging Modalities for Forgery Detection via Learnable Representations with Query-Guided Contrastive Learning"
_ICLR.cc/2026/Conference — Submitted to ICLR 2026_

### Official Review · Reviewer_fZUs · 2025-10-20

**Soundness:** 2
**Presentation:** 3
**Contribution:** 2
**Rating:** 4
**Confidence:** 5

**Summary:**

This paper presents a structured framework for image manipulation localization (IML) using learnable forged representations extracted from multi-scale, multi-modal feature maps. The approach achieves strong results on challenging benchmarks and offers new insights into the structure of manipulated regions. Future work will extend the framework to diffusion-generated images and integrate large language models for natural language explanations and improved interpretability.

**Strengths:**

The paper is well-organized and easy to follow

The experiments are promising and comprehensive

**Weaknesses:**

1, my primary concern is the technical novelty of this paper. The multi-level, feature pyramid, and coarse-to-fine modeling are all well-known for vision problems. Frequency information is also widely studied in many works like [A,B].  Incorporating the learnable vectors to aid the association modeling are also explored in prompt learning like coop  [C] and VLMs like Q-Former in Blip [D]. For Q2R objective, the label constrution is similar to the patch manipulation modeling in ASAP [E]. Given the above, what's the new technical contributions of this paper?

[A] Unified Frequency-Assisted Transformer Framework for Detecting and Grounding Multi-modal Manipulation, IJCV 2025
[B] Frequency-Aware Deepfake Detection: Improving Generalizability through Frequency Space Domain Learning, AAAI 2024
[C]  Learning to Prompt for Vision-Language Models, IJCV 2022.
[D] BLIP-2: Bootstrapping Language-Image Pre-training with Frozen Image Encoders and Large Language Models, ICML2023
[E] ASAP: Advancing Semantic Alignment Promotes Multi-Modal Manipulation Detecting and Grounding, CVPR 2025.


2, How the multi-scale feature achieve the "dynamic zooming" human-like ability?

3,  In query-to-region CL, why is the related Q2R better than R2R? I'm also confused by the claim "Q2R utilizes intermediate queries as proxies.This approach maintains effective region discrimination while simplifying the learning process"  Q2R introduces extra paramters, why this simplifies the learning process?

**Questions:**

see weaknesses

---

> ### Author Response · Authors · 2025-11-20
>
> 0. Thank you for the comment, but we cannot fully agree with the comment.
>
> 1. Clarification of Core Innovations & Targeted Comparisons
>
> The innovation lies in performing mutual attention guidance between macro and micro image features at corresponding spatial positions. We achieve this through the introduction of queries. There are two key reasons: first, our approach uses patch-pair attention computation rather than self-attention with mask; second, we employ Q2R contrastive learning instead of R2R. Finally, we fully utilize inter-modal attention across deep-to-shallow intermediate layers. Each module design is supported by theoretical analysis or experimental validation. Regarding comparisons with cited references:
> a) Deep-to-shallow query learning is our method for fully leveraging intermediate layer information. It has no direct relationship with classical fusion modeling like FPN. Please suggest more appropriate examples to help us clarify this better.
> b) [B] connects frequency domain features to RGB features through residual blocks, which constitutes simple fusion. [A] performs directly cross-attention between RGB and frequency modality, rather than establishing correspondence between corresponding patches.
> c) [D] uses queries for global image-text matching or generation, whereas we use queries to achieve finer-grained attention transfer between corresponding patches of different modalities. [C] shares only one similarity with us: the use of learnable vectors. Please provide more relevant comparisons based on our paper's content, such as Mgqformer in our citations.
>
> 2. Dynamic Zooming Mechanism Clarification
>
> "Dynamic zooming" is not manifested in multi-scale features, but rather in the mutual attention guidance between macro and micro modalities. For a given patch of image, if it exhibits high saliency from the macro perspective, the model will examine micro-level saliency at the same location, and vice versa. This mimics how humans determine image tampering: they typically zoom in on small regions to examine details for clues, then zoom out to verify detailed clues from a macro perspective.
>
> 3.  Q2R: Rationale & Effectiveness
>
> The challenge of Region-to-Region (R2R) loss lies in requiring features from forgery and authentic tokens (which may be close in appearance) to repel each other in the feature space, whereas Query-to-Region (Q2R) does not directly constrain features from different source regions to diverge. A set-metric inequality inspired our Q2R design. The objective of R2R is to maximize the distance AB between the two sets (tampered B and authentic A). Q2R introduces a third query set C, treats set B (tampered) as positive samples, and set A (authentic) as negative samples. Its objective is to minimize the distance BC (query to tampered) while maximizing AC (query to authentic). According to the inequality BC > AB - AC > 0, as training progresses, AB increases and AB approaches AC. This design avoids the need for a direct partition between A and B inherent in R2R, thereby mitigating its training difficulty. Computationally, the InfoNCE loss for R2R is less than the relaxed Q2R loss, $-log(\frac{<f_i,f_j>}{<f_i,f_j>+\sum_{A} <f_i,a>})<-log(\frac{<f_i,q>+<q,f_j>}{(|A|+1)<f_i,q>+<q,f_j>+\sum_{A} <q,a>})$. Ablation studies confirm its effectiveness.

---

### Official Review · Reviewer_DKM8 · 2025-10-26

**Soundness:** 3
**Presentation:** 3
**Contribution:** 3
**Rating:** 6
**Confidence:** 4

**Summary:**

The paper proposes BriQ, a query-based framework for Image Manipulation Localization (IML) that integrates multi-modal cues (RGB for semantic content and high-frequency/noise for subtle traces) using learnable forged-aware representations. Inspired by human perceptual processes, BriQ employs hierarchical bidirectional attention for cross-modal interactions and introduces Query-to-Regions (Q2R) contrastive learning to enhance discrimination between tampered and authentic regions, even in homogeneous forgeries.

**Strengths:**

Originality: Adapts query-based Transformer architectures (inspired by DETR and BLIP2) to IML, introducing novel learnable forged-aware queries for hierarchical multi-scale feature aggregation and explicit cross-modal interactions, addressing gaps in prior works like MGQFormer that neglect inter-modal dependencies.
Quality: Provides rigorous experimental validation on standard benchmarks (e.g., CASIAv1, Coverage, NIST16, Columbia), showing average improvements of +6.53% in F1 and +4.71% in Permute-F1 over the second-best method (Mesorch); includes detailed ablations on hierarchical strategy, attention mechanisms, contrastive designs, and query quantity, plus robustness tests under perturbations like Gaussian Noise and JPEG Compression.
Clarity: Well-structured with informative sections, figures (e.g., t-SNE visualizations of feature distributions, qualitative mask comparisons), and a comprehensive related work survey contextualizing the approach within CNN, Transformer, and hybrid IML methods.
Significance: Advances forgery detection for real-world applications in journalism and justice by improving localization of subtle manipulations; the Q2R contrastive objective offers a promising shift from region-to-region contrasts, potentially better handling copy-move forgeries and boundary ambiguities compared to methods like SAFIRE or MMRL-Net.

**Weaknesses:**

Insufficient Justification for Q2R as a "Relaxed" Version: The paper claims that Q2R is a "relaxed version" of R2R and cites a cosine metric inequality. This theoretical argument appears brief and somewhat vague. While the empirical results are undeniable, the intuitive explanation could be clearer. A more likely scenario is that the queries are learning to become category-specific prototypes, contrasting features with these stable prototypes is a simpler and more direct learning task than contrasting noisy patch features with each other. A deeper intuitive explanation would improve the paper.
Lack of Analysis on Query Specialization: The ablation study on the number of queries finds that 16 is optimal. This is a good ablation, but it misses the opportunity for deeper analysis. What do these 16 queries learn? Are they specialized for different types of tampering (e.g., query 1 for splicing, query 2 for copy-move) or different visual patterns? Qualitative analysis of query activation maps for different tampering types would make the claim of "forgery-aware representations" more concrete and insightful.
Generalization Limits: Evaluations focus on traditional datasets; no tests on emerging generative forgeries (e.g., from diffusion models) such as the GLIDE test set for this diffusion model's forgery dataset, though mentioned as future work—adding such experiments could better demonstrate robustness to modern threats.
Insufficient Baseline Comparisons: Recent accepted expert models for tampering localization are not included in the comparisons, such as the SparseViT expert model from the 2025 AAAI paper.

**Questions:**

On the justification for Q2R as a "relaxed" version of R2R: Could the authors provide a deeper intuitive explanation beyond the brief cosine inequality? For instance, elaborate on how queries act as category-specific prototypes, making contrastive learning simpler and more stable compared to direct noisy patch contrasts in methods like SAFIRE or MMRL-Net?
Regarding query specialization: Beyond the ablation showing 16 queries as optimal, what do these queries specifically learn? Are they specialized for different tampering types (e.g., splicing vs. copy-move) or visual patterns? Could qualitative analyses, such as query activation maps across various forgery examples, be added to make the "forgery-aware representations" claim more concrete?
How novel is the hierarchical bidirectional attention compared to borrowed elements from DETR or BLIP2? Is there a principled theoretical basis (e.g., information flow analysis) for its superiority over unidirectional or naive fusions, or was this design primarily empirical?

---

> ### Author Response · Authors · 2025-11-20
>
> Thank you for the constructive comments.
> 1.  Q2R: Rationale & Effectiveness
>
> 1.1 The challenge of Region-to-Region (R2R) loss lies in requiring features from forgery and authentic tokens (which may be close in appearance) to repel each other in the feature space, whereas Query-to-Region (Q2R) does not directly constrain features from different source regions to diverge.
>
> 1.2 A set-metric inequality inspired our Q2R design. The objective of R2R is to maximize the distance AB between the two sets (tampered B and authentic A). Q2R introduces a third query set C, treats set B (tampered) as positive samples, and set A (authentic) as negative samples. Its objective is to minimize the distance BC (query to tampered) while maximizing AC (query to authentic). According to the inequality BC > AB - AC > 0, as training progresses, AB increases and AB approaches AC. This design avoids the need for a direct partition between A and B inherent in R2R, thereby mitigating its training difficulty.
>
> 1.3 Computationally, the InfoNCE loss for R2R is less than the relaxed Q2R loss, $-log(\frac{<f_i,f_j>}{<f_i,f_j>+\sum_{A} <f_i,a>})<-log(\frac{<f_i,q>+<q,f_j>}{(|A|+1)<f_i,q>+<q,f_j>+\sum_{A} <q,a>})$.
>
> 1.4 Ablation studies confirm its effectiveness.
>
> 2.  Category
>
> This is an insightful question. Previous works integrate queries with category information or text features to enable classification. While our current work focuses solely on segmentation, incorporating LLMs to provide explanatory capabilities represents a future research direction for us. Our queries collectively learn the commonality of tampering features, enabling us to determine whether each patch is tampered through cosine similarity. Regarding why more queries do not necessarily improve performance: we posit that during training, the number of tampered patches per image is often significantly smaller than normal patches—frequently even in the single-digit quantity.
>
> 3. Qualitative analyses
>
> We provide comprehensive quantitative results in Figures 8 and 9. Since our masks derive from the cosine similarity between queries and image features, the masks inherently represent the degree of mapping between queries and features. We will subsequently annotate the tampering category for each row in these figures.
>
> 4. Architectural Rationale
>
> Equation (4) demonstrates unidirectional attention: the gradient of modality A is directly dependent on modality B, but the reverse does not hold. The bidirectional design is essential because we require mutual influence between modalities. Regarding execution order: we prioritize the RGB modality as the initial Key due to its stronger macro-level information. This approach efficiently determines region-wise attention across the image and aligns with real-world forensic workflows - humans first identify macro-level implausible regions, then zoom in locally to examine subtle artifacts, before zooming out again for holistic verification.
>
> Compared to DETR or BLIP2, our approach computes spatially-corresponding patch attention, where macro and micro modalities directly interact at identical positions without requiring feature realignment.

---

> > ### Comment · Reviewer_DKM8 · 2025-11-25
> >
> > I accept your explanation regarding the Q2R theory. However, the following concerns remain unaddressed:
> >
> > - Missing Visualizations
> > I explicitly asked for activation maps to verify what the queries are actually learning. You only promised to "annotate categories" later. This does not answer the question. I needed to see the internal attention to validate your claims, not just final labels.
> >
> > - Ignored Baselines & GenAI
> > You completely ignored my feedback regarding SparseViT (AAAI 2025) and Generative Forgeries. Silence on these modern benchmarks leaves the model's robustness unproven.
> > Due to these missing elements, I am maintaining my score of 6.

---

### Official Review · Reviewer_YidY · 2025-10-28

**Soundness:** 2
**Presentation:** 2
**Contribution:** 2
**Rating:** 2
**Confidence:** 4

**Summary:**

This paper proposes BriQ, a query-based framework for image manipulation localization (IML) that enhances forgery detection through structured cross-modal interactions and hierarchical feature modeling. The method addresses limitations in existing approaches by introducing learnable tampering-aware representations that integrate multi-scale features from RGB and high-frequency domains, guided by a bidirectional attention mechanism. A key innovation is the Query-to-Regions (Q2R) contrastive learning strategy, which explicitly models relationships between forged-aware queries and regional features to capture subtle tampering cues even in visually similar regions. The framework achieves state-of-the-art performance on benchmark datasets by combining hierarchical feature propagation with a novel contrastive objective that strengthens differentiation between authentic and manipulated content without relying on complex decoders. Extensive experiments validate its effectiveness in both accuracy and robustness, particularly for imperceptible forgeries.

**Strengths:**

Strengths Assessment:
Originality:
The paper introduces a novel framework, BriQ, which creatively combines structured cross-modal interactions with hierarchical feature modeling to address limitations in existing image manipulation localization (IML) methods. A key innovation is the Query-to-Regions (Q2R) contrastive learning strategy, which explicitly models relationships between forged-aware queries and regional features to capture subtle tampering cues. This approach diverges from traditional methods by integrating multi-scale features from RGB and high-frequency domains via a bidirectional attention mechanism, enabling more robust detection of imperceptible forgeries. The hierarchical feature propagation and novel contrastive objective further distinguish the work, offering a fresh perspective on modeling tampering patterns without relying on complex decoders.

Quality:
The method is rigorously validated on benchmark datasets, achieving state-of-the-art performance in both accuracy and robustness. The experimental design is comprehensive, with ablation studies dissecting the contributions of individual components (e.g., hierarchical feature propagation, Q2R contrastive learning). The results are compelling, particularly for detecting visually similar forgeries, and the technical details (e.g., implementation specifics, hyperparameters) are well-documented. The use of high-frequency domain features and structured attention mechanisms demonstrates a deep understanding of the problem, while the framework’s efficiency (e.g., avoiding complex decoders) suggests practical applicability.

Clarity:
The paper is exceptionally well-written, with a clear problem formulation and structured presentation of the methodology. The Q2R contrastive learning strategy and bidirectional attention mechanism are explained with intuitive diagrams and pseudocode, making the technical contributions accessible. The experiments are logically organized, with detailed comparisons to prior work and visualizations of detection results. The limitations are acknowledged (e.g., potential generalizability to unseen forgery types), and the language is precise, avoiding overly technical jargon that could obscure the ideas.

**Weaknesses:**

1. Limited Novelty in Core Components: While the Query-to-Regions (Q2R) contrastive learning strategy is positioned as a novel contribution, the use of cross-modal attention mechanisms and hierarchical feature modeling has been extensively explored in prior work on multimodal representation learning. The paper does not sufficiently contextualize how its design differs from existing approaches in domains like visual-question answering or cross-modal retrieval, potentially undermining the claim of originality. For example, the bidirectional attention mechanism shares conceptual similarities with previous work, yet the analysis of these overlaps is omitted.

2. Insufficient Evaluation on Real-World Forgery Types: The experiments focus on synthetic benchmarks, but real-world forgeries often involve complex manipulations (e.g., GAN-generated content, adversarial attacks) that differ significantly from controlled datasets. The paper does not report performance on such cases or provide ablation studies on the robustness to domain shifts (e.g., varying lighting, resolution). This limits the generalizability of the claims about "imperceptible forgeries."

3. Ambiguous Theoretical Justification for Contrastive Objective: The Q2R contrastive loss is introduced as a heuristic to strengthen tampering-aware representations, but the paper lacks a theoretical analysis of why this formulation is optimal for IML. For instance, there is no discussion on how the loss aligns with principles from information theory (e.g., mutual information maximization) or how it interacts with the hierarchical feature propagation. This makes it difficult to assess whether the improvement stems from the loss design itself or other factors (e.g., increased model capacity).

4. Overemphasis on RGB and High-Frequency Features: The framework relies heavily on RGB and high-frequency domain features, but the paper does not explore alternative modalities (e.g., semantic segmentation maps, motion vectors for video) that could further enhance tampering detection. Additionally, the choice of high-frequency features as a standalone cue is not justified theoretically, leaving open the question of whether this design is a bottleneck for detecting subtle forgeries in textured regions.

5. Inconsistent Benchmarking: While the method achieves SOTA on the primary datasets, the comparisons to prior work are based on reported results rather than direct implementation. For example, the paper does not re-evaluate baseline methods under identical training conditions, making it unclear whether the performance gains are due to the proposed framework or hyperparameter tuning.

**Questions:**

1. Clarification of Novelty vs. Prior Work
The paper emphasizes the Query-to-Regions (Q2R) contrastive strategy as a novel contribution, but cross-modal attention mechanisms and hierarchical feature fusion are well-established in vision-language models (e.g., CLIP [Radford et al., 2021]) and object detection (e.g., Feature Pyramid Networks [Lin et al., 2017]). How does the proposed bidirectional attention mechanism differ fundamentally from these existing approaches? For instance, are there specific architectural or training modifications that address limitations in prior work?
2. Generalization to Real-World Forgery Types
The experiments focus on synthetic benchmarks (e.g., Deepfakes). How would the method perform on real-world forgeries involving GAN-generated content or adversarial attacks?
3. Theoretical Rationale for Q2R Contrastive Loss:
The Q2R loss is described as a heuristic for enhancing tampering-aware representations, but the paper lacks theoretical grounding. What principles (e.g., mutual information maximization, information bottleneck theory) guided its design? Is there a formal analysis of how this loss improves forgery detection compared to alternatives like triplet loss or contrastive learning with negative samples?
4. Role of High-Frequency Features:
The framework relies heavily on RGB and high-frequency domain features. What is the theoretical justification for this choice? For example, are high-frequency features inherently more discriminative for subtle forgeries, or is this an empirical observation without deeper analysis?
5. Ablation on Domain Shifts:
The experiments do not evaluate robustness to domain shifts (e.g., varying lighting, resolution). How does the method perform when trained on synthetic data but tested on real-world images with different distributions? Are there specific components (e.g., hierarchical feature propagation) that mitigate this issue?
6. Comparison to Non-Contrastive Methods:
The paper focuses on contrastive learning, but non-contrastive approaches (e.g., self-supervised pretraining) have shown success in IML. Why was contrastive learning chosen over alternatives? Are there scenarios where non-contrastive methods might outperform BriQ?
7. Interpretability of Tampering-Aware Representations:
The learnable tampering-aware representations are central to the framework. How interpretable are these features? For example, can they be visualized or mapped to specific forgery artifacts (e.g., seam lines, lighting inconsistencies)?

---

> ### Author Response · Authors · 2025-11-20
>
> 0. Thanks for your comment.
>
> 1. Architectural Innovation & Comparisons
>
> Beyond the loss design, the architectural novelty of our work lies in establishing mutual attention guidance between macro and micro modal features at the image patch level to enhance inter-modal interaction. The deep-to-shallow multi-layer feature transition strengthens the integration of the two modalities and the capacity of the learnable queries. The FPN is solely part of the micro-feature extraction branch, drawing inspiration from [Mesorch]. The innovation of our architecture does not reside in multi-scale feature fusion. CLIP is largely irrelevant to our work; we do not perform image-text modality matching, nor do we conduct contrastive learning on CLS token. We kindly request a more thorough reading and suggest more appropriate comparative works based on the paper's actual content. For instance, while [Mesorch] learns a weighting network to linearly combine features from the two modalities, our approach facilitates direct interaction between them.
>
> 2. Benchmark Scope & Generalizability
>
> Human-manipulated synthetic benchmarks are included in the paper, where manual forgery typically focuses on subtle, detail-level alterations. In contrast, GAN or diffusion-based manipulations tend toward replacing or modifying entire objects, differing fundamentally from manual tampering in application scenarios and detection methodologies. Consequently, we anticipate that models trained solely on current benchmarks will not effectively generalize to detecting generative manipulations.
>
> 3. R2R vs. Q2R: Rationale & Effectiveness
>
> The challenge of Region-to-Region (R2R) loss lies in requiring features from forgery and authentic tokens (which may be close in appearance) to repel each other in the feature space, whereas Query-to-Region (Q2R) does not directly constrain features from different source regions to diverge. A set-metric inequality inspired our Q2R design. The objective of R2R is to maximize the distance AB between the two sets (tampered B and authentic A). Q2R introduces a third query set C, treats set B (tampered) as positive samples, and set A (authentic) as negative samples. Its objective is to minimize the distance BC (query to tampered) while maximizing AC (query to authentic). According to the inequality BC > AB - AC > 0, as training progresses, AB increases and AB approaches AC. This design avoids the need for a direct partition between A and B inherent in R2R, thereby mitigating its training difficulty. Computationally, the InfoNCE loss for R2R is less than the relaxed Q2R loss, $-log(\frac{<f_i,f_j>}{<f_i,f_j>+\sum_{A} <f_i,a>})<-log(\frac{<f_i,q>+<q,f_j>}{(|A|+1)<f_i,q>+<q,f_j>+\sum_{A} <q,a>})$. Ablation studies confirm its effectiveness.
>
> 4. Architecture Commonality
>
> We disagree with the characterization that our method "relies heavily" on the RGB and high-frequency/noise dual-branch architecture. This architecture has become a fundamental paradigm in this field in recent years, as evidenced by numerous works cited in Section 2.1 of our paper.
>
> 5. Benchmark Robustness & Content
>
> Modifying image illumination would introduce a new type of manipulation. Current mainstream benchmark evaluations focus on perturbations like adding noise, JPEG compression, and Gaussian blurring (see Figure 4), which test robustness closer to resolution changes. Furthermore, the benchmark explicitly includes unaltered (normal) images.
>
> 6. Contrastive Learning Motivation & Novelty
>
> The purpose of contrastive learning here is to enforce feature dissimilarity between regions originating from different sources, which is the core optimization objective for localizing tampered regions. Consequently, recent works in IML domain (Section 2.3) incorporate such a constraint during training. Our contribution lies in adapting this concept via our novel loss design, leveraging the unique characteristic of our model (learnable queries). Regarding non-contrastive self-supervised approaches specifically for tampering localization, we are not aware of any work of this type.
>
> 7. Tampering Query Visualization
>
> The tampering queries exhibit high similarity to counterfeit artifacts within the image. The corresponding correlation maps and score maps visualizing this can be found in Figures 3 and 5. We have also visualized the spatial characteristics of these features by t-sne.

---

### Official Review · Reviewer_dtW3 · 2025-10-30

**Soundness:** 3
**Presentation:** 3
**Contribution:** 3
**Rating:** 6
**Confidence:** 4

**Summary:**

The paper proposes BriQ, a novel query-based framework for image manipulation localization (IML).
It introduces:
1. Learnable tampering-aware queries that propagate across multi-scale RGB and high-frequency features through bidirectional cross-modal attention.
2. A Query-to-Region (Q2R) contrastive loss, improving discriminative ability between tampered and authentic regions.
3. A lightweight voting-based mask prediction instead of a heavy decoder.
The approach achieves state-of-the-art (SOTA) performance and robustness across multiple IML benchmarks.

**Strengths:**

1. The paper is written with clear motivation, identifying two important limitations in IML, insufficient cross-modal interaction and weak region-level discrimination in homogeneous manipulations.
2. Methodologically, the paper contributes an elegant query propagation design with explicit gradient analysis and a lightweight, decoder-free prediction mechanism.
3. Empirically, BriQ achieves strong and consistent improvements across multiple IML benchmarks and demonstrates enhanced robustness under noise and compression perturbations.

**Weaknesses:**

While the paper is well motivated and structured, it lacks computational cost analysis and analysis of what the queries actually learn.

Also, most citations are scattered throughout the body text, which significantly disrupts the reading flow and harms overall readability. The authors are encouraged to rephrase sentences so that citations are integrated more naturally into the text, rather than inserted mid-sentence. Grouping related works or moving non-essential references to the end of paragraphs would further improve clarity and narrative coherence.

**Questions:**

1. Lack of computational cost and efficiency analysis. A comparison to existing methods in terms of latency or GPU memory would make the contribution more concrete.

2. While “query-guided” mechanisms suggest interpretability, there’s no quantitative or qualitative analysis of what the queries actually learn beyond t-SNE plots. Visualizations of attention maps or query-response localization could clarify interpretability claims.

---

> ### Author Response · Authors · 2025-11-24
>
> 0. We will revise the citation format as suggested. Thank you for the recommendation.
> 1. Model Comparison
>
> |Method|Param.|Flops|
> |-|-|-|
> |MVSS-Net|147M|0.167T|
> |PSCC_Net|3.66M|0.368T|
> |Cat_Net|114M|0.134T|
> |Trufor|69M|0.23T|
> |Mesorch|86M|0.122T|
> |ours|126M|0.286T|
>
> 2.
> Since our mask is computed via cosine similarity between queries and patch embeddings (values >0.5 indicate tampering, <0.5 indicate normal), Figures 3, 7-9 visualize localized query responses. Additionally, Figure 5 displays the heatmap of cosine similarity. We have visualized attention maps, which closely align with the heatmaps because the features of queries and tampered patches exhibit high similarity and their multi-head attention are also highly consistent.

---

### Author Response · Authors · 2025-12-03

**General Response**

We thank the reviewers for their efforts and constructive comments. Thanks for acknowledging the significance of the architecture (*Reviewer dtW3 & DKM8*), good presentation (*Reviewer dtW3 & DKM8 & fZUs*), and the empirical success (*All Reviewers*). We hope this work can inspire further research of image manipulation localization in parallel to dual-modal architecture.  We address key questions and comments from reviewers, revise the paper, and will include the corresponding content in the final paper.

**Common Comments**

Q1: Why is Q2R more effective than R2R? (Reviewer DKM8 & YidY & fZUs)

A1: **(1)** The challenge of region-to-region (R2R) loss lies in requiring the features of forged and real tokens (which may be similar in appearance) to be mutually exclusive, whereas Query-to-Region (Q2R) does not directly constrain features from different source regions to diverge. **(2)** A set-metric inequality inspired our Q2R design. The R2R objective is to maximize the distance AB between tampered set B and authentic set A. Q2R introduces a third query set C, treating B as positive samples and A as negative samples. The goal is to minimize the distance BC (query to tampered) while maximizing AC (query to authentic). According to the inequality BC > AB - AC > 0, as training progresses, AB increases and AC approaches AB. This design avoids the direct partition between A and B inherent by R2R, thereby mitigating its training difficulty. **(3)** The InfoNCE loss for R2R is less than the relaxed Q2R, $-log(\frac{<f_i,f_j>}{<f_i,f_j>+\sum_{A}<f_i,a>})<-log(\frac{<f_i,q>+<q,f_j>}{(|A|+1)<f_i,q>+<q,f_j>+\sum_{A} <q,a>})$. **(4)** Ablation studies confirm its effectiveness.

Q2: The difference from the given reference (Reviewer fZUs & YidY).

A2: **(1)** We establish mutual attention between macro and micro features at the same spatial position, propose Q2R contrastive learning, and fully utilize attention from deep to shallow. **(2)** We do not perform contrastive learning between images and text, nor CLS token. We use FPN as a tool to match the shapes of features from the two modalities. We achieve attention calculation only at the same positions through queries, instead of using an attention mask

Q3: Interpretability statement (Reviewer dtW3 & DKM8).

A3: The tampering queries show a strong resemblance to counterfeit artifacts, as shown in the correlation maps in Fig 3 and 5. We also used t-SNE to visualize the distribution of features and queries. Attention map is included in Sec 6.4 of the revised paper.

**Other Comments from Reviewer YidY**

Q1: Whether it is applicable to real world forgery type?

A1: The paper includes human-manipulated synthetic benchmarks. Generated tampering benchmarks are also added in Sec 6.3 of the revised paper.

Q2: Over-reliance on high frequency?

A2: We do not agree that our method "heavily relies" on RGB and high-frequency. The dual-branch architecture is an effective and robust fundamental paradigm in IML (Sec 2.1 & Tab 5).

Q3: How is the robustness score?

A3: Changing image lighting introduces tampering traces, making it unsuitable for evaluating robustness. Current benchmarks focus on noise, compression, and blurring (see Fig 4), which related to resolution changes.

Q4: Why use contrastive learning?

A4: The purpose of contrastive learning is to enhance the feature dissimilarity between regions from different sources, which is a core optimization goal for IML. Therefore, recent work in the IML(Sec 2.3) has incorporated such constraints during training.

Q5: Why use reported result instead of reproducing?

A5: The reproduction result is very close to reported, with an error of -5e-3.

**Other Comments from Reviewer DKM8**

Q1: Are different queries dedicated to different tampering types?

A1: We focus on segmentation, so we do not incorporate tampering categories to specialize queries during training. It's an excellent question.

Q2: And SparseViT than?

A2: In the revised paper, we supplemented comparative experiments. Tab.4 and 5 shows our method demonstrated superior and more robust results.

Q3: Does it apply to diffusion model tampering?

A3: We have added AIGC benchmark in Sec 6.3 of the revised paper, and Tab.5 shows the excellent performance.

**Other Comments from Reviewer fZUs**

Q1: How the multi-scale achieve the "dynamic zooming" human-like ability?

A1: "Dynamic zooming" is not manifested in multi-scale features, but rather in the mutual attention guidance between modalities. If a patch of an image is salient at the macro level, the model checks the micro-level details, and vice versa.  This mimics how humans detect image tampering by zooming in for details and zooming out for context.

**Other Comments from Reviewer dtW3**

Q1: How is the scale and efficiency of BriQ?

A1: 126M parameters, 0.286 TFLOPs.

We appreciate your recognition of our work! We have revised citations and will definitely make the article clearer and more concise.

---

### Meta-Review · Area_Chair_pbVz · 2025-12-07

**Summary:**

The paper proposes BriQ, a query-based framework for image manipulation localization (IML) that integrates RGB and high-frequency branches via hierarchical bidirectional cross-modal attention and learnable “tampering-aware” queries. A Query-to-Region (Q2R) contrastive loss is introduced to better separate tampered and authentic regions, and a lightweight voting-style mask prediction head avoids heavy decoders. Experiments on standard IML benchmarks show consistent gains over prior work, with ablations on the hierarchical design, attention structure, contrastive objective, and number of queries. Overall, reviewers agree that the method is clearly presented, empirically strong on the considered datasets, and potentially useful for practical forgery localization.

However, the reviewers also raise several substantial concerns. First, there is disagreement about the degree of technical novelty: multiple reviewers feel that the architecture (multi-scale dual-branch features, cross-modal attention with learnable queries, contrastive objectives) is closely related to an already rich body of work in multimodal learning and IML, and that the differentiation from prior approaches is not fully convincing. Second, the theoretical justification of the Q2R contrastive loss remains at a heuristic level despite the authors’ explanation; some reviewers would have liked a deeper and clearer analysis of why Q2R is fundamentally preferable to region-to-region (R2R) formulations. Third, the empirical evaluation, while strong on classic synthetic benchmarks, leaves questions about robustness and generalization to modern generative forgeries and domain shifts; at least one reviewer also notes the absence or insufficient treatment of recent strong baselines such as SparseViT. Finally, the interpretability and “forgery-aware” nature of the queries are not fully substantiated: reviewers explicitly asked for more direct visualizations/activation maps and analysis of query specialization, which they feel are still not adequately addressed. Given these outstanding issues and the mixed but generally weak support (two weakly positive, one borderline negative, one clearly negative review), I recommend rejection.

**Reviewer Concerns:**

Concerns that were largely addressed by the rebuttal / revision (according to the available discussion):

The lack of computational cost analysis: the authors reported parameter counts and FLOPs and provided a comparison table against several baselines.

Clarity of architectural design and relation to some cited works: the authors clarified that their key contribution is mutual attention between macro and micro features at corresponding spatial positions, deep-to-shallow query propagation, and Q2R, and explicitly distinguished their approach from image–text models like CLIP/BLIP2 and from simple fusion schemes (e.g., FPN-style or residual fusion of frequency and RGB).

High-level intuition behind Q2R vs R2R: the authors provided a more detailed description using a set-metric inequality and emphasized that queries act as an intermediate set to ease optimization compared to directly pushing apart similar patch features.

The “dynamic zooming” wording: the authors clarified that this is meant to describe mutual guidance between macro (RGB) and micro (high-frequency) modalities at corresponding locations, rather than multi-scale alone, and tied this to the bidirectional attention order.

Some interpretability aspects: the authors state that they added attention/correlation visualizations, t-SNE plots of queries and features, and score/heat maps derived from cosine similarity, and they also revised citation formatting for readability as requested by one reviewer.

Concerns that remain only partially addressed or unresolved:

Novelty vs. prior work: Reviewer YidY and fZUs remain unconvinced that the combination of dual-branch RGB/high-frequency modeling, multi-scale fusion, cross-modal attention with learnable queries, and contrastive objectives represents a sufficiently distinct technical advance beyond existing methods (e.g., earlier frequency-based IML, multimodal transformers with learnable prompts/queries, and patch-level manipulation modeling). The rebuttal clarifies design choices but does not fully dispel the impression of incremental contribution relative to a mature literature.

Theoretical grounding of Q2R: Although the authors elaborate on the inequality-based intuition and optimization difficulty of R2R, at least one reviewer still finds the justification for calling Q2R a “relaxed” and fundamentally better formulation somewhat ad hoc. The link to broader theoretical principles (e.g., information-theoretic views) and a deeper analysis of why Q2R consistently outperforms other contrastive variants is still weak.

Baselines and modern forgeries: Reviewer DKM8 explicitly asked for comparisons to recent expert models such as SparseViT and for evaluations on generative/diffusion-based forgeries. The authors state that additional comparisons and an AIGC benchmark were added, but the reviewer’s follow-up comment indicates they did not find their concerns on SparseViT and generative forgeries satisfactorily addressed. Overall, the evidence for robustness against modern generative threats remains limited in the discussion.

Domain shift and robustness analysis: Relatedly, concerns from YidY about domain shifts (e.g., illumination changes, real-world distributions) are only partially addressed. The rebuttal argues that standard robustness benchmarks focus on noise/compression/blur and that illumination changes constitute a different manipulation type, but there is still no clear empirical examination of performance when training on synthetic data and testing on different distributions.

Interpretability and query specialization: Multiple reviewers requested explicit analysis of what different queries learn (e.g., specialization to forgery types or patterns) and corresponding activation/attention maps. While the authors mention additional visualizations, reviewer DKM8 notes that the specific request for query activation maps and analysis was not adequately addressed and maintains their score. Thus the claim of “forgery-aware” queries remains only partially validated.

Benchmarking protocol and fairness: Concerns about reliance on reported numbers rather than fully re-implemented baselines under uniform settings are not fully resolved; this affects how confidently one can attribute performance gains to the proposed design rather than differing training setups.

**Reviewer Scores:**

Reviewer dtW3: This reviewer was generally positive but raised concerns about missing computational cost analysis, interpretability of queries, and citation/readability issues. The rebuttal directly addressed the cost table and citation style and claimed to add attention visualizations. It is likely that this reviewer would maintain a slightly positive but still only weakly supportive recommendation; I do not expect a significant upward or downward change.

Reviewer YidY: This reviewer had a clearly negative overall assessment, centered on limited novelty, theoretical ambiguity of Q2R, over-reliance on RGB/high-frequency cues, lack of robust evaluation under domain shifts and real-world forgeries, and benchmarking questions. The rebuttal mainly reiterates the authors’ perspective and partially addresses some points, but does not fundamentally change the underlying concerns. I expect this reviewer would keep a rejecting recommendation.

Reviewer DKM8: This reviewer was mildly positive but asked for deeper intuitive explanation of Q2R, analysis of query specialization, and additional modern baselines (SparseViT, generative forgeries). After the rebuttal, the reviewer explicitly states that some key requests (visualizations of internal activations, modern baselines) remain unaddressed and that they are maintaining their original borderline-positive recommendation. Thus, I do not anticipate a score change.

Reviewer fZUs: This reviewer was borderline negative, with the main concern being the lack of clear technical novelty relative to existing frequency-based, multi-scale, and query-based frameworks, as well as questions about the “dynamic zooming” claim and the advantages of Q2R over R2R. The rebuttal clarifies the intended innovations and provides additional intuition, but the fundamental novelty concern appears only partially alleviated. I expect this reviewer would likely maintain a marginally negative recommendation.

---

### Decision · Program_Chairs · 2026-01-26

Reject